# Co-Designing Digital Technologies for Improving Clinical Care in People with Parkinson’s Disease: What Did We Learn?

**DOI:** 10.3390/s23104957

**Published:** 2023-05-22

**Authors:** Mariana H. G. Monje, Sylvie Grosjean, Martin Srp, Laura Antunes, Raquel Bouça-Machado, Ricardo Cacho, Sergio Domínguez, John Inocentes, Timothy Lynch, Argyri Tsakanika, Dimitrios Fotiadis, George Rigas, Evžen Růžička, Joaquim Ferreira, Angelo Antonini, Norberto Malpica, Tiago Mestre, Álvaro Sánchez-Ferro

**Affiliations:** 1HM CINAC, Hospital Universitario HM Puerta del Sur, Universidad CEU-San Pablo, 28968 Madrid, Spain; mariana.hgmonje@gmail.com; 2Ken and Ruth Davee Department of Neurology, Northwestern University Feinberg School of Medicine, Chicago, IL 60611, USA; 3Department of Communication, Com&Tech Innovations Lab (CTI-Lab), University of Ottawa, Ottawa, ON K1N 6N5, Canada; 4Department of Neurology and Centre of Clinical Neuroscience, First Faculty of Medicine, Charles University and General University Hospital, 128 21 Prague, Czech Republic; 5CNS—Campus Neurológico, 28933 Torres Vedras, Portugal; 6LAIMBIO, Laboratorio de Análisis de Imagen Médica y Biometría, Universidad Rey Juan Carlos, 2560-280 Madrid, Spain; 7Dublin Neurological Institute, Mater Misericordiae University Hospital, D07 W7XF Dublin, Ireland; 8PD Neurotechnology, 15451 Athens, Greece; 9Parkinson and Movement Disorders Unit, Department of Neurosciences (DNS), Padova University, 35131 Padova, Italy; 10Parkinson’s Disease and Movement Disorders Center, Division of Neurology, Department of Medicine, The Ottawa Hospital Research Institute, The University of Ottawa Brain and Research Institute, Ottawa, ON 60611, Canada; 11Movement Disorders Unit, Neurology Department, Hospital Universitario 12 de Octubre, 28041 Madrid, Spain

**Keywords:** digital technologies, codesign, digital care, Parkinson’s disease

## Abstract

The healthcare model is shifting towards integrated care approaches. This new model requires patients to be more closely involved. The iCARE-PD project aims to address this need by developing a technology-enabled, home-based, and community-centered integrated care paradigm. A central part of this project is the codesign process of the model of care, exemplified by the active participation of patients in the design and iterative evaluation of three sensor-based technological solutions. We proposed a codesign methodology used for testing the usability and acceptability of these digital technologies and present initial results for one of them, *MooVeo.* Our results show the usefulness of this approach in testing the usability and acceptability as well as the opportunity to incorporate patients’ feedback into the development. This initiative will hopefully help other groups incorporate a similar codesign approach and develop tools that are well adapted to patients’ and care teams’ needs.

## 1. Introduction

Parkinson’s disease (PD) is a neurodegenerative condition that, for optimal management, may benefit from the implementation of multidisciplinary and personalized care strategies [1]. With digital technologies, this multispecialty care can be re-shaped to reach more people with PD in their homes and communities. Digital technologies also have the potential to connect patients with the care team beyond the typically sparse clinical visits while helping with the remote monitoring of the disease and fostering care continuity.

The ease of use or the social acceptability of digital tools may increase the interest in using these solutions [2]. Therefore, to successfully navigate the transformation of healthcare towards integrated care approaches, it is essential to involve patients more closely in the design of care plans [3]. The iCARE-PD project aims to address this need by developing a technology-enabled, home-based, and community-centered integrated care model [4]. The technology-enabled care initiative piloted in iCARE-PD comprises three sensor-based self-tracking tools enabling patients/caregivers to monitor disease manifestations, treatment compliance and response, and record health information. A central pillar of this project is the codesign process of the model of care, which makes patients and care partners active participants in the design and iterative evaluation of technological solutions [5,6].

Usability and acceptability have yet to be systematically tested in PD technologies, despite the significant expansion of the field and the undoubted information we could obtain with its use [7]. The codesign of technological solutions would help guarantee the target users’ usability and acceptability of the digital technologies. The main objective of this study is to evaluate the usability and acceptability of three digital technologies according to user-friendliness (ease of use), user confidence and user satisfaction.

In this paper, we start by proposing a mixed-methods methodology to operationalize the codesign process, testing the usability and acceptability of digital solutions. To evaluate this methodological approach, we tested it in one of the digital technologies developed by our Consortium.

## 2. Methods and Analysis

### 2.1. Study Design

We are currently conducting a multicenter-international study in five tertiary PD centers (Fundación Investigación HM Hospitales, Spain; Ottawa Hospital Research Institute, Canada; Charles University, Czech Republic; CNS—Campus Neurológico, Portugal; and Mater Misericordiae University Hospital Dublin, Ireland). In this study, we aim to test the following three digital technologies developed by the iCARE-PD Consortium: *MooVeo* [8]; *SpiroGym* [9] and *PDMonitor^®^* [10]. We plan to enroll a total of 40 People with Parkinson’s Disease (PwP).

### 2.2. Study Sample and Recruitment

PwP were enrolled at the clinical sites based on these criteria: (1) PD diagnosis according to the MDS clinical diagnostic criteria [11]; (2) willing and able to sign informed consent and complete the questionnaires and study assessments. A sample size of 30 participants would allow detecting 95% of possible user errors for an estimated probability of occurrence of 0.15 [12].

### 2.3. Experimental Set-Up

This was a 2-week study, composed of a screening/baseline visit, and a final visit. The study procedures are detailed in Figure 1. After having obtained the patient’s consent, the investigator collected demographic, medical, and medication history. Clinical data includes the time of diagnosis, most affected side, Hoehn and Yahr stage and the MDS-UPDRS part III in the “ON” state. Study subjects tested the digital technologies as follows: *MooVeo* on site at visit 1 and *SpiroGym* and *PDMonitor^®^* at home for 1 week between visit 1 and visit 2. It took around 10 to 15 min to demonstrate the *MooVeo* technology to the participants and address their queries.

Both the usability and acceptability of the three digital technologies were evaluated after their use with the following questionnaires: SUS (System Usability Survey) [13] and an ad hoc iCARE-PD questionnaire (example in Appendix A). The SUS is a short, reliable tool for measuring the usability of a system. It consists of a 10-item questionnaire with five response options for respondents; from Strongly agree to Strongly disagree. The final score ranges from 0 to 100, corresponding to a percentile ranking. SUS scores above 68 are considered above average [13]. To operationalize the codesign process, we developed the iCARE-PD questionnaire to evaluate the patients’ perspective on the envisioned use of each technology [14,15,16]. The questionnaire is composed of two parts. Part 1 addresses the acceptability and usefulness of digital health technologies on the basis of 14 questions with five Likert-type response options from Strongly agree to Strongly disagree and a free-text response to complete the story about the use of the digital technology. Part 2 addresses the user perspective on the application of digital technology for self-care in daily life, and consists of a list of nine paired words capturing different attitudes towards the use of digital technology, and the subjects choose the most fitting (e.g., engaging vs. stressful). Finally, a section with open-answer questions for recommendations to codesign the digital technology is provided. The questionnaires were self-administered via an online survey using the REDCap platform. The participants completed the questionnaires between visit 1 and visit 2. To limit the possible site variability and to guarantee the uniformity of assessment across the centers: (1) trained PD professionals conducted the training in the different types of technology; (2) we excluded patients who could not communicate independently to control variance due to those factors; and (3) the *MooVeo* platform and the questionnaires were conducted in the official language of the site’s country to control variance through a language barrier.

Integrated Care Digital Technologies Evaluated:-*MooVeo* is a software package designed to help physicians and patients with PD to track their disease manifestations remotely using the webcam of a standard computer [8]. The patient stands in front of the computer at a specific distance and runs *MooVeo.* Through text and figures, the software guides the patient through three simple motion tasks, detailing how to perform them, and records videos of the different tasks. When the patient is in front of the camera, the software localizes different points on the hand (the fingers) of the patient. Therefore, when the patient performs the task, the software “follows” the hand and measures the movement. On the basis of this, the software generates various metrics (i.e., mean amplitude or speed of the movement), which are used to generate a report that can be sent to the patient or the care team. This software can be used by the patient to monitor his/her motor conditions. Additionally, the neurologist can potentially monitor changes related to treatment and symptom progression, and it can even help in diagnosis. The software can be run locally or as a cloud application, with recorded videos being uploaded to the cloud and automatically quantified in a secure HIPAA/GDPR-compliant manner.-*SpiroGym* is a mobile phone application designed to help patients increase their self-management, motivation and adherence to a respiratory physiotherapy program [9]. Patients train their respiratory strength with the assistance of a commercially available expiratory muscle trainer device and an externally added microphone. The microphone captures the expiratory sound during respiratory training and the SpiroGym app transforms it into a graph. The app thus gives patients visual feedback about the quality of their use of the expiratory muscle trainer device. The SpiroGym app also allows the patient to check on training data from previous workouts and therefore monitor long-term development [9].-*PD Monitor* is a non-invasive continuous monitoring system for use by PD, certified as a Class IIa Medical Device according to European regulation EE 93/42/EEC [10]. It is composed of a set of wearable devices, a mobile application that enables patients/caregivers to record medication, nutrition, self-assessed motor and non-motor status information, and a physician tool, which graphically presents all patient-related information.

### 2.4. Data Management

De-identified clinical data were collected and managed using REDCap electronic data capture in compliance with the GDPR (General Data Protection Regulation) and Personal Information Protection and Electronic Documents Act (PIPEDA) data protection regulations. A unique identification code was assigned to each participant. The results of the present study for one of the technologies, *MooVeo*, is provided in the next section.

## 3. Results

A total of 31 patients with PD used *MooVeo*. A summary of demographic and clinical characteristics is provided in Table 1.

### 3.1. Usability of MooVeo

The mean average SUS score of the participants was 73, ranging from 30 to 100. The evaluation of the individual items showed that a high percentage of the users felt confident using the tool (90%) and thought it was easy to use (86%). A total of 79% of the participants would either like to use it in the future or felt neutral about it. The proportions of responses to each question are detailed in Figure 2.

### 3.2. Acceptability of MooVeo

The patients reported good acceptability and appreciated the usefulness of *MooVeo* (Figure 3). A total of 89% of participants thought the technology was pleasant to interact with and easy to learn to use, while 82% felt they could use it independently without someone else’s help. Moreover, 79% felt they would use the technology again, and 66% agreed that the information provided by *MooVeo* would help manage their condition. On the other hand, 45% considered it an acceptable method for self-management (Figure 3). Regarding user perspectives, 70% thought they could integrate it into their daily routine, while 50% could imagine themself using *MooVeo* in the future. The report generated by *MooVeo* was easy to understand and interpret for 33% of the respondents. In comparison, 53% felt the graph and visual feedback would help them better manage their condition at home.

### 3.3. Codesign Suggestions

In Appendix A, we report some of the relevant suggestions that the participants provided to further improve the technology. Relevant quotes are organized according to four overarching themes.

## 4. Discussion

In this study, we presented a mixed-methods approach that can be used to codesign digital sensor-based technologies with patients in order to test their usability and acceptability. We analyzed the initial results for 31 patients using one of the evaluated technologies, *MooVeo*.

*MooVeo* was perceived overall to have higher-than-average usability (73 on the SUS scale, which represents a 70% percentile ranking and is considered to be above-average usability as measured with this instrument) and was considered to be user friendly by the majority of investigated subjects (86%). Despite the user-friendliness, patients were unclear about whether they would routinely use it in the future to monitor their disease, based on the SUS responses (Figure 2). When we analyzed the acceptability of the technology, a high percentage of the participants (89%) found it pleasant to interact with it, and most participants were able to use it independently and would be keen on using it again (Figure 3).

Two main key areas for improvement that we identified during the codesign process were related to: (i) the perceived help the patient was receiving from the technology for self-care of PD; and (ii) the visualization of the results. These two areas for improvement are probably interrelated, as the capacity to self-manage the disease is linked with how the information is provided to the user [17]. Additionally, the participants requested that the technology better explain what is intended, so they can understand the clinical utility, and the possibility of “gamification” would make the experience more fun, which could be particularly useful for addressing some of the challenges envisioned with continuous home use (Appendix A).

### 4.1. Lessons Learned When Codesigning Digital Technologies with Patients in an International Consortium

As we stated in the introduction, there is a need for the patient-centered development of technologies to allow the remote assessment of patients’ conditions in their natural home environment, promoting a more comprehensive clinical evaluation and empowering patients to monitor their disease. Overall, the literature has oversimplified self-care, leading to the design of individualized technologies that are probably not able to fit with the complexity of the different dimensions of the self-care of PD patients [18,19].

Co-care implies a shift from episodic routine-driven care to more flexible care management, driven by the mutual needs of patients and heath care professionals. Previous initiatives have aimed to explore how co-care could be operationalized in PD care, supported by eHealth. For example, it has been shown that individual constraints include eHealth literacy and acceptance [20]. This is of critical relevance, as it could be very informative in the early stages of the design and development of solutions.

Another critical aspect of the development of technologies in PD is to engage patients in its use. To achieve this, digital technologies need to be perceived as both realistic and feasible by users. Therefore, it is critical to include users’ perspectives and experiences in the development process. This area is frequently overlooked when developing digital medical technologies. In our study, we identified that the patient report was not always understood, and this will allow us to refine what information to include and how to make a user-friendly representation of the data generated by *MooVeo*. By exploring the possibilities of designing individual tailored visualizations representing patient-generated data, we will facilitate patients’ understanding when reviewing their personally generated data [21].

Overall, this point is also important because the design of this technology needs to ensure engagement and effective use in real life, and if we had not followed this codesign approach we would probably have learned this later in the research and development (R&D) process, with likely implications in terms of needing to repeat some of the development steps.

The other challenging and revealing area was the use of multiple technologies in a codesign study. We had to balance the use of each technology and the burden of the included procedures. This is relevant because implementing objective technological measures in day-to-day clinical practice is a significant challenge. Technology integration in PD care needs to be safe, effective, patient-centered, timely, efficient, equitable and secure. Thus, the compliance and feasibility for the users is key for its use and the continuous development of digital technologies and how they will integrate with each other also needs to be considered carefully.

Furthermore, the technologies were at different stages of maturity, with two of the systems still having lower technology readiness levels (TRLs < 6 for *MooVeo* and *SpiroGym*), while one other was a fully approved product (TRL 9 for *PDMonitor^®^*) [22]. Despite this, our approach seemed informative at all stages of technology readiness. Therefore, we highly encourage including these analyses in any technological R&D project, and not just focusing on the clinical/diagnostic performance of the developed solution.

### 4.2. Limitations

The cohort reported here is an adequate representation of a PD population, but the study sample size is admittedly not large; thus, the generalizability of the results to the broader PD population (e.g., including patients at more advanced disease stages and different medications) may need to be revised. However, the international nature of the consortium and a larger-than-usual sample size for usability studies partially alleviate this risk [12]. On the other hand, we tested the use of this technology *(MooVeo*) at the hospital, so that further research evaluating the acceptability of this technology for its use in a less controlled environment, such as the home environment, will be necessary.

Furthermore, the use of the technologies was restricted in time (one-time use for *MooVeo*, and a week for the other technologies, *SpiroGym* and *PDMonitor^®^*), and to patients who agreed to be part of the study, Thus, in future studies, we will assess usability and technology acceptance during a long-term use period in a less controlled setting and using a broader population of patients. Additionally, it will be interesting to ascertain the usability and acceptability of the technology to neurologists/treating physicians, and how it will change the management of PD patients.

Other areas of research, besides improving the above aspects, include completing the study (N = 40 participants), evaluating the other two technologies that were also included in the study, and demonstrating in clinical utility studies the value of an integrated care model supported by these types of digital solutions.

## 5. Conclusions

The involvement of PD patients in the design of digital tools results in a shift to a more flexible approach to care delivery based on the mutual needs of both patients and care professionals. Here, we demonstrated the usefulness of this approach for one of the technologies, *MooVeo*, that was developed by the iCARE PD Consortium. We demonstrated *MooVeo’s* usability and user acceptance, as well as several areas for improvement concerning the self-management of PD and the notification system. This initiative will hopefully help other groups to incorporate a similar codesign approach early in the R&D process and develop tools adapted to patients’ and care teams’ needs. We will continue using this approximation with the other technologies we are currently testing in the iCARE-PD Consortium towards the development of a digitally supported integrated PD care model.

## Figures and Tables

**Figure 1 sensors-23-04957-f001:**
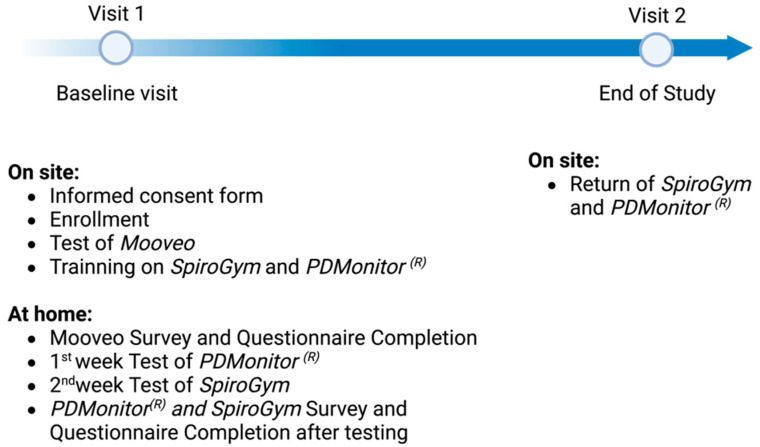
Representation of the study flowchart.

**Figure 2 sensors-23-04957-f002:**
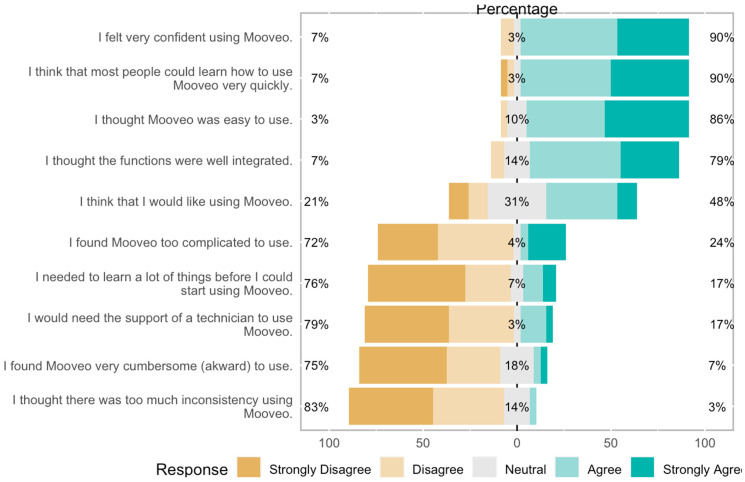
Individual items of SUS evaluation for the MooVeo platform.

**Figure 3 sensors-23-04957-f003:**
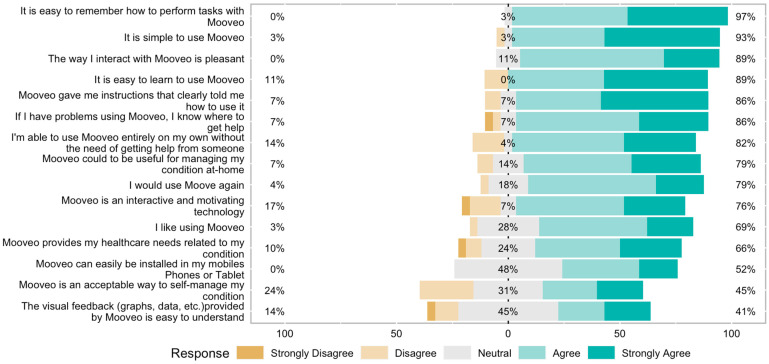
Codesign survey responses.

**Table 1 sensors-23-04957-t001:** Baseline characteristics of the PD patients.

Baseline Characteristics		
Age (years)		66.5 (48.5–82.5)
Sex (female/male)	Male	19 (61.3%)
	Female	12 (38.7%)
Disease duration (years)		8 (1.64–25.6)
Hoehn and Yahr stage	1. Symptoms on one side only	2 (6.5%)
	2. Symptoms on both sides but no impairment of balance	23 (74.2%)
	3. Balance impairment. Mild to moderate disease	5 (16.1%)
	4. Severe disability, but able to walk or stand unassisted	1 (3.2%)
MDS-UPDRS III		27.5 (5–119)

Values are median (interquartile range) MDS-UPDRS III (the Movement Disorder Society-Sponsored Revision of the Unified Parkinson’s Disease Rating Scale, part III) scores.

## Data Availability

The data are available on request to qualified investigators.

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
