# Peer review of "Co-Designing Digital Technologies for Improving Clinical Care in People with Parkinson’s Disease: What Did We Learn?"

_sensors, 2023, doi:10.3390/s23104957_

Round 1

Reviewer 1 Report

Comments:

The article is about co-designing digital technologies for improving the patient care in Parkinson’s disease. The research question is novel and clinically relevant. The study is an ongoing multicenter study and the authors present the usability and acceptability of  one of the digital technology (MooVeo) used in their study. The study design is adequate and the manuscript is well written. Some points which needs to be addressed:

1-     The technology MooVeo was used at the baseline visit. There is no mention of time taken to demonstrate the technology to the participants and address their queries regarding it. What was the average time spent per participants?

2-     The assessment for the usability and applicability of MooVeo was done at same visit (i.e baseline visit). Assessment of usability and acceptability may be misleading if it is done immediately after demonstration or training. It should have been done after a gap of few days and after patients have used the technologies a minimum number of times.

3-     As the training for the technology were done in different centers, did the authors noted any difference between the sites ? How do the authors address the site variability ?  What measures were taken to maintain uniformity of assessment across the centers?

4-     How much time the participants took for answering the questionnaires? Were the questionnaires self-administered?

5-     The sample size is very small for evaluating the three technologies in a multicentic trial. How the sample size was calculated ?

6-     Figure 1 mentions that survey and questionnaire completion was done at visit 1 and also at visit 2. Which questionnaire was administered at which time should be clarified in the manuscript.

7-     The authors mention that 33% respondents found the report generated by the MooVeo was easy to undertstand. Authors should provide a sample of the report as supplementary material.

8-     One week is a short period to assess the acceptability of a new technology. Long term study should be required.

9-     It would be interesting to know the usability and acceptability of the technology by the neurologist/treating physicians and how it changed the management.

Reviewer 2 Report

Dear Authors,

thank you for giving me the possibility to review your paper. It was very good to read and I found it very interesting. It's crucial nowadays to enable best healthcare for more than 10 million people with PD worldwide. As PD is a disease that includes movement and non-movement symptoms, an appropriate and individualized approach and therapy plays a key role in its treatment. As it affects mainly older people, they often need care partners to support them. In the Internet of Things era, with fast developing digitalization and Medical Internet the subject of the presented paper is very relevant. Moreover, sensors and applications allowing for remote health condition tracking will play an important role in the healthcare systems in the near future. With fast growing human population and aging population, health monitoring, or diagnosing patients remotely may contribute to some relief of the burden on the health care system and may reduce waiting times to see a specialist. On the other hand, the described solutions will improve the work of doctors, using Medical Internet will enable even faster diagnosis and allow the implementation of appropriate treatment and its modifications based on the data recorded by the patient's sensors.

Although, I find the paper very interesting and its results promising, text requires reediting. There is different font size within the text e.g., in lines 21-23, 37-41, 81, 287. Moreover, double spacing occurs in the text and some brackets are missing. Please use same citation style and reference manager.

Looking forward to your further research,

Reviewer

Reviewer 3 Report

The paper proposes a field study on the adoption and perception of a few technologies for patients affected with Parkinson's disease.

The work is solid and brings interesting results in terms of perception of technology among patients with a permanent disease. However the topic itself seems only remotely connected to the journal interest. 

Besides, the study does not, in my opinion, go far enough and no result is really unexpected. Concerning acceptation, for example, the figures are highly positive, but the sample is based on patients who agreed to be part of the study, and I am therefore unsure of the possibility to generalize the findings to the general population of patients. 

On another dimension, it would be interesting to also have results on SpiroGym and PD Monitor, and to see long term series to compare the evolution of the disease with and without these helper tools. 

Round 2

Reviewer 1 Report

The authors have addressed my previous queries adequately and have made necessary changes in the manuscript. 

Reviewer 3 Report

I understand that the study is still preliminar and that additional results and more patients should be included in future works. My comments were therefore not addressed, but I still think the paper does present an interesting scenario and for that reason I would not block it. 

My only concern is the appropriateness of the topic vs. the editorial line of the journal. I think that other journals could have been a better fit.